# Effectiveness and cost-effectiveness of a sustainable obesity prevention programme for preschool children delivered at scale 'HENRY' (Health, Exercise, Nutrition for the Really Young): protocol for the HENRY III cluster randomised controlled trial

Maria Bryant [ORCID],[1,2] Wendy Burton,[2] Michelle Collinson,[3] Adam Martin,[4] Bethan Copsey [ORCID],[3] Dawn Groves-Williams [ORCID],[3] Alexis Foster [ORCID],[5] Thomas A Willis,[6] Philip Garnett,[7] Alicia O'Cathain [ORCID][5]

For numbered affiliations see end of article.

**Correspondence to**
Professor Maria Bryant;
maria.bryant@york.ac.uk

## ABSTRACT

**Introduction** One-fifth of children start school already overweight or living with obesity, with rates disproportionately impacting those living in the most deprived areas. Social, environmental and biological factors contribute to excess weight gain and programmes delivered in early years settings aim to support families to navigate these in order to prevent obesity. One of these programmes (Health, Exercise and Nutrition for the Really Young, HENRY) has been delivered in UK community venues (hereon named 'centres') in high deprivation areas since 2008 and aims to help families to provide a healthy start for their preschool children. We aim to establish the effectiveness and cost-effectiveness of HENRY, including its potential role from a wider systems perspective.

**Methods and analysis** This is a multicentre, open-labelled, two-group, prospective, cluster randomised controlled trial, with cost-effectiveness analysis, systems-based process evaluation and internal pilot. Primary analysis will compare body mass index (BMI) z-score at 12 months in children (n=984) whose parents have attended HENRY to those who have not attended. Secondary outcomes include parent and staff BMI and waist circumference, parenting efficacy, feeding, eating habits, quality of life, resource use and medium term (3 years) BMI z-scores (child and siblings). 82 centres in ~14 local authority areas will be randomised (1:1) to receive HENRY or continue with standard practice. Intention-to-treat analysis will compare outcomes using mixed effects linear regression. Economic evaluation will estimate a within-trial calculation of cost-per unit change in BMI z-score and longer-term trajectories to determine lifelong cost savings (long-term outcomes). A systems process evaluation will explore whether (and how) implementation of HENRY impacts (and is impacted by) the early years obesity system. An established parent advisory group will support delivery and dissemination.

## STRENGTHS AND LIMITATIONS OF THIS STUDY

⇒ The Health, Exercise and Nutrition for the Really Young (HENRY) III trial provides an opportunity to examine the effectiveness and cost-effectiveness of an intervention that has been publicly funded and delivered at scale for more than 10 years, including short-term, medium-term and long-term outcomes.

⇒ Our systems-based process evaluation will explore the role that community-based obesity prevention interventions play within the wider system, based on the understanding that factors that influence obesity are complex and multifaceted.

⇒ The HENRY III trial has been designed following a successful feasibility study, in collaboration with local and national government partners, members of the public and a multidisciplinary trial team.

⇒ The trial's success is dependent on recruitment of local authority areas, where additional funding will be required to commission the intervention.

⇒ A mitigation strategy has been developed to reduce the risk of selection bias inherent in cluster designs.

**Ethics and dissemination** Ethical approval has been granted by the University of York, Health Sciences' Research Governance Committee (HSRGC/2022/537/E). Dissemination includes policy reports, community resources, social media and academic outputs.

**Trial registration number** ISRCTN16529380.

## INTRODUCTION

Reducing the prevalence of childhood obesity among people living in deprived areas is a public policy priority. Approximately one-third of children (27.7% on average and up to 34.5% in higher

deprivation areas) are defined as overweight (13.3%) or having obesity (14.4%). Health inequalities have also broadened, from a 6.3% difference in rates of overweight and obesity between the most and least deprived areas in 2019/2020 to 10.7% in 2021/2022.[1 2]

Preventing excess weight in childhood is beneficial for health and well-being, as well as reducing medium-term to long-term burden on health services for those living with obesity, including mental health services and those used to treat and manage respiratory diseases such as asthma.[3 4] Obesity prevention during childhood can also reduce excess weight gain and obesity in later life[5 6] which is difficult to reverse once established[7–9] and is a cause of numerous long-term, chronic health conditions. However, interventions to prevent obesity generally result in modest but inconsistent benefits.[7–10] Given that obesity is caused by a wide range of physiological, psychological, environmental, economic and social factors, this is unsurprising and means that the role and cost-effectiveness of locally delivered programmes within a large and complex socioeconomic and public health system is uncertain. Public Health England advocated systems approaches; encouraging local areas to adopt a range of interconnecting interventions and policies inside and outside the healthcare sector to collectively tackle obesity and related health inequalities.[11] It remains unknown what role individual interventions play in disrupting this system,[12] and which of these interventions are most effective and cost-effective. Locally delivered public health prevention programmes can be cost-effective[13]; however, there is limited evidence specifically looking at obesity prevention delivered at scale. Further, while there has been an emergence of evidence that explores the role that understanding the obesity system has for adults[14–16] since the Foresight map was published,[17] there is a lack of understanding of what an early years obesity system looks like and what the implications of understanding this might be.

One childhood obesity prevention programme which has been delivered at scale for many years in the UK is HENRY (Health, Exercise and Nutrition for the Really Young); a community-based programme, designed to alter early years settings, upskill the early years workforce and improve lifestyle behaviours of parents/carers (hereon called 'parents') and their preschool aged (under 5 years old) children. Although HENRY was designed to be a universal programme, it has been predominantly delivered in children's centres/community venues located in areas of high deprivation. Evidence suggests it has potential to impact on population obesity[18] but its effectiveness is not yet established. The current trial adopts a novel approach to evaluate the effectiveness and cost-effectiveness of the programme and explore the potential role of HENRY in disrupting the multiple factors that influence excess weight gain in the system. This novel approach places an emphasis on exploring if and how HENRY influences the system in which it operates (including system balancers which may reduce its potential for population impact).

Following on from a successful feasibility study,[18] our current evaluation will consider child obesity outcomes in the short-term (using data collected in the trial), medium-term (using routinely collected data at 3 years) and longer-term (using economic modelling and secondary datasets).

## AIMS AND OBJECTIVES
### Aim
To establish the effectiveness and cost-effectiveness of an obesity prevention programme delivered at scale, including its potential role from a wider systems perspective.

### Primary objective of randomised controlled trial
The primary objective of the trial is to determine whether HENRY reduces child age-adjusted and sex-adjusted body mass index (BMI) at 12 months.

### Secondary objectives of randomised controlled trial
1. To determine whether HENRY improves parent self-efficacy, eating behaviours, feeding behaviours, dental health and quality of life.
2. To explore whether HENRY influences rates of obesity in parents, siblings and staff (health practitioners).
3. To examine the social and physical environment in the children's centres.
4. To monitor any safety issues from the intervention, including adverse events or unintended consequences

### Internal pilot objectives of randomised controlled trial
To assess centre recruitment, parent recruitment and HENRY programme delivery against predefined progression criteria.

### Economic evaluation objectives
1. To explore the long-term effects on child BMI.
2. To determine whether HENRY provides an overall cost saving (eg, to the NHS).

### Process evaluation objectives
1. To produce a map of the system within which HENRY operates and identify hypotheses about how this may be disrupted in response to HENRY.
2. To analyse the system in which HENRY is embedded to understand how the system and its elements change over time in response to HENRY.
3. To undertake a traditional process evaluation nested within the randomised controlled trial (RCT) to understand reach of HENRY within target population, potential contamination and how it has been implemented.

## METHODS AND ANALYSIS
### Trial design
A multicentre, open-labelled, two-group, prospective, cluster RCT, with cost-effectiveness analysis and embedded

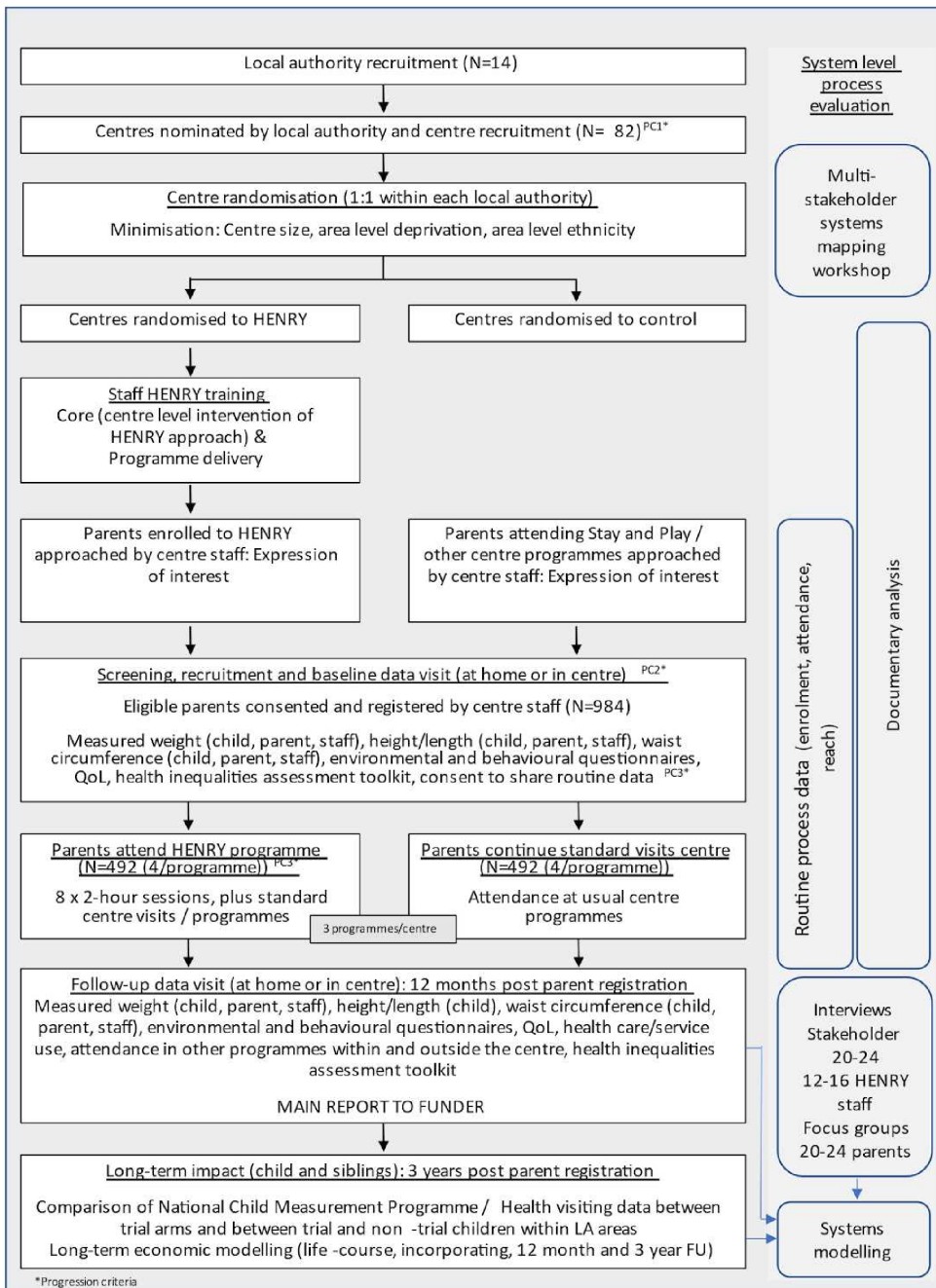

**Figure 1** Trial summary. FU, follow-up; HENRY, Health, Exercise and Nutrition for the Really Young; QoL, quality of life.

mixed methods complex systems evaluation and internal pilot (figure 1). 82 eligible children's centres from within ~14 local authorities (depending on the number of centres per local authority) will be randomly allocated (1:1) to deliver HENRY or continue with standard practice (control). 984 eligible parents will be recruited.

### Setting and recruitment
#### Local authority and centre recruitment
We plan to recruit local authorities or other associations (eg, NHS Trusts, private organisations) that are willing to commission HENRY across the UK, from which centres and parents will be recruited (figure 2). In addition to inviting areas who actively express an interest in

the intervention (independently from the trial), we will promote the trial in partnership with all National Institute of Health Research (NIHR) Clinical Research Networks (CRN) in England and the equivalent organisations in the devolved nations. Eligible areas (criteria are detailed below) will then be asked to (1) nominate at least two (ideally six or more) centres which meet trial eligibility criteria and (2) sign an agreement before entering the trial. We will record reasons for declining participation, and basic demographic information, from those authorities that choose not to participate.

Local authorities will be asked to nominate approximately twice the number of centres for which they

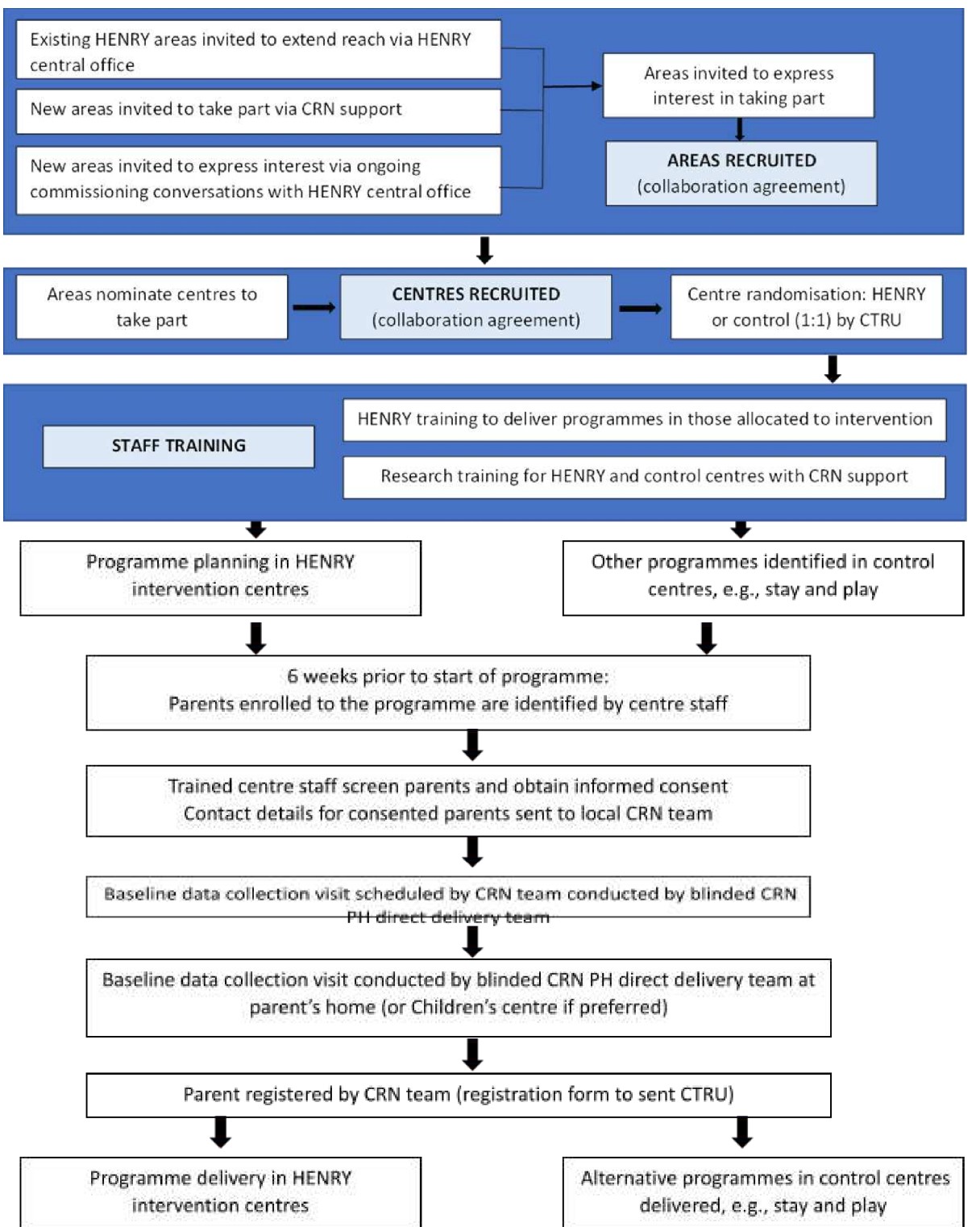

**Figure 2** Flow of recruitment. CRN, Clinical Research Networks; CTRU, Clinical Trials Research Unit; HENRY, Health, Exercise and Nutrition for the Really Young.

wish to commission HENRY, so that approximately half can be randomised to receive HENRY and half can be randomised to control. Although there are no exclusions based on the demographics of centres, location will be monitored and commissioners encouraged to nominate centres to include a range of diverse social and environmental characteristics. Centre managers will be given information about the research and asked to provide consent for their centre to participate through a collaboration agreement.

### Parent/carer and staff recruitment
Centre staff will invite all parents who are booked to attend a HENRY programme to take part in the research

(and will be remunerated per participant based on NIHR CRN service support costs[19]). If interested, they will be screened for eligibility and will be asked to provide informed consent to take part. Those who decline will still be able to attend the HENRY intervention if they wish. Screening and recruitment will aim to start at least 6 weeks prior to the start of each HENRY programme. Programmes typically enrol an average of eight parents, of whom our design aims to recruit an average of four parents. Once consented, blinded data collection staff will contact participants to arrange the baseline visit and complete trial registration. A similar process will recruit parents from the control centres. Here, parents who attend other programmes (eg, stay and play sessions) will be invited to take part within the same time frames. Parents will also have the opportunity to self-refer into the trial via recruitment posters displayed in the centres. A £30 shopping voucher will be offered to participants in recognition of their contribution (£15 at baseline, £15 at follow-up). Parent consent will include five options: consent to take part in the trial only, consent to provide height and weight measurements, consent to provide HENRY programme attendance data (if attending a HENRY programme), consent to be contacted about taking part in a process evaluation interview and consent to share routinely collected National Child Measurement Programme (NCMP) data after 3 years for the purposes of the economic evaluation.

Staff recruitment will occur once they have completed site training. Those agreeing to take part will provide their height, weight and waist circumference, which they can self-measure. Staff will also have the opportunity to consent to be contacted about taking part in a process evaluation interview.

### Eligibility criteria

Local authorities can be new to HENRY or already commissioning HENRY, provided they have at least two centres (ideally six) meeting centre eligibility criteria. Local authorities using external teams outside of the centre to deliver HENRY programmes (eg, health visitors) will be eligible, in addition to those wishing to train internal centre staff to deliver programmes (the most common model). Local authorities without coverage of a NIHR local CRN (LCRN; the teams responsible for collecting trial data from parents and staff) will not be eligible.

Centres: Any type of centre or other early years setting such as a nursery or community venue will be eligible to take part provided that they do not already deliver HENRY, and their staff have not received HENRY training (within the past 2 years). This includes both public and private nurseries. Centres where staff are shared between nominated centres will not be eligible. Centres must aim to run HENRY programmes starting within 12 weeks of training completion and plan to run three programmes during their trial participation period (approximately 18 months). They should be in geographically separate areas to protect against contamination (judged on a case by case basis) and managers must agree to support participant recruitment within their centres. If the centre operates as part of a cluster, that cluster must be deemed to be HENRY naive. HENRY naive clusters are defined as a group of centres within a cluster that does not include any centres that are (a) currently delivering HENRY or (b) have been trained to, or delivered HENRY within the past 2 years.

Centres that are either currently delivering HENRY or have previously delivered HENRY in the last 2 years are not eligible.

Parents/carers: The target population for the intervention is parents/carers of preschool children. Parents may not be registered more than once but they may be screened on more than one occasion if not registered following first screening, as both eligibility and willingness to participate may change. Parents must have at least 1 child aged 6 months to 5 years (18 months to 6 years at 12-month follow-up) at the time of starting the programme. If more than one child in a family fulfils eligibility criteria, the youngest child (by birth timing if twins) will be considered as the reference child (from which data will be collected). Parents must be willing to attend the programme sessions (intervention centres) and willing to provide data in accordance with the data collection protocol. Parents will be provided with full details of the data collection requirements in advance so that they can make informed decisions as to whether to participate. They must either speak English or bring their own interpreter with them (eg, family member) each time they need to respond to questionnaires for trial purposes. Where available, non-English-speaking parents will also be able to use local interpreters (provided by the centre or through the LCRN). Parents with severe learning difficulties that preclude them taking part in group sessions in which they need to be able to read and write will not be eligible. Those whose reference child is tube fed or with other known clinical conditions likely to affect growth over the period of the trial (eg, coeliac disease) and those who have attended a HENRY group for a previous child will also be ineligible.

We have developed a recruitment strategy based on learning from our feasibility study[18] which includes screening and consenting by centre staff (who are familiar to families) and detailed training for centre staff, including clear timelines and expectations. For example, in order to meet targets, a minimum of four parents should be recruited within a 6-week window prior to the start of each HENRY programme (and at an equivalent time in control centres).

All centre staff will be invited to participate in the research including those directly and indirectly involved in delivering HENRY programmes.

### Randomisation and blinding

Following fully signed local authority (and service provider if applicable) and centre agreements,

participating centres within each local authority will be randomised to HENRY or control in a 1:1 allocation ratio by the Leeds Clinical Trials Research Unit. Minimisation, incorporating a random element, will be used to ensure the treatment groups are well balanced for the following characteristics:

► Size of centre (≤8/>8 permanent centre members of staff, not including staff using the centre such as Health Visitors, nursery workers, etc).
► Area-level ethnicity (<80%/≥80% white British using Census data based on centre postcode).
► Area-level deprivation (≤10%/>10% ranking within Index of Multiple Deprivation at the lower layer super output area).

Parents and centre staff will not be blinded due to the nature of the intervention. Data collection staff will be blinded to centre allocation (and thus the participants' treatment arm).

### Active intervention

HENRY was set up with Department of Health funding in 2008 and has been widely commissioned by more than 50 areas (training >15 000 practitioners and providing programmes to ~24 500 parents). It includes core practitioner training and group facilitation training (www.henry.org.uk).[20]

HENRY is delivered primarily within Sure Start centres and other children's centres, as well as other community settings, including schools, mosques and churches. Local authority areas can choose to have HENRY programmes delivered directly by HENRY central teams (less common approach), pay to commission training, licensing and support from HENRY (most common approach), offer a blended approach or provide training to local staff to deliver programmes (most common in larger local authority areas).

### Intervention training for practitioners

Core practitioner training supports staff to deliver the HENRY approach, which incorporates evidence-based behaviour change models, including the family partnership model, motivational interviewing and solution-focused support, with information about a healthy start that is consistent with national guidance. This training is designed to allow staff to integrate evidence-based models to develop motivation and support lifestyle change for families. This can then be embedded into all interactions with families, in addition to supporting positive changes to the centre environment (space to play, freely available water, food policies, etc). Training is provided to health visitors, dieticians and staff (eg, at children's centres, community centres/hubs—hereon called 'centres') allowing parental support to be an intrinsic part of their role, while influencing culture and policy within early years settings. Training is designed to consistently influence the environment and practice immediately following training.

Facilitator training is delivered to a (usually) smaller selection of staff who have attended core training to certify them to deliver small group sessions 'Right from the Start'. This training enables the delivery of an 8-week universal 'Right from the Start' programme to parents to provide practical skills in authoritative parenting skills, increasing self-esteem, adopting healthy family lifestyles, goal setting, oral health, active play, portion sizes and learning about food labels.

Training can be delivered in-person or remotely, depending on commissioner and facilitator preference. To allow a richer level of communication, the in-person training can be completed in a shorter time frame (2-day in-person training is delivered over 6 weeks remotely).

### Programme delivery to parents

The 'Right from the Start' programme is delivered to groups of approximately 8–10 parents over 8 sessions. Each interactive session focuses on a separate theme and includes resources for families to take home. Each centre delivers 2–3 'Right from the Start' group programmes per year, each consisting of 8×2 hours sessions.

Programmes can be delivered remotely or face to face. We will include both delivery modes in the proposed research. This will allow us to be flexible depending on external issues, in addition to supporting an evaluation of a pragmatic intervention. Our analysis plan includes consideration of different delivery approaches.

Parents with a child aged up to 5 years are eligible to attend HENRY. The programme can be tailored to specific groups or parents, for example, to align with dietary practices for different ethnicities or religion. Language needs are addressed locally depending on the population needs, with some areas providing dedicated support. Without this, the other approach is to invite parents to attend with a friend or family member to support translation and other activities. Although considered to be a universal programme, practitioners often refer families 'at risk' to HENRY and the delivery model within children's centres allows parents from the most deprived neighbourhoods to attend.

Parents receive a standardised manual as part of the programme. To monitor adherence, session attendance will be recorded for parents who consent.

### Control intervention

Staff within control centres will not receive HENRY training and will continue to deliver usual programmes or 'standard practice' (eg, 'stay and play', 'cook and eat'). In short, this means that families registered to take part in the research will receive the standard level of support provided within their community/centres. Services are aimed at supporting families with a focus on the most disadvantaged families. These vary between and within local areas but usually include access to health visiting teams, breastfeeding support, parenting advice and access to specialist services including speech and language therapy.

**Table 1** Outcome data collection summary (see online supplemental table 1) for a full list of data and outcomes

| Outcomes | Measures | Collected by | Participant/ staff baseline | Short-term follow-up (12 months postregistration) | Medium-term follow-up (3 years)* | Longer-term follow-up† |
|---|---|---|---|---|---|---|
| Child age-adjusted and sex-adjusted BMI Z-score, child height, child weight, unadjusted BMI and weight/BMI percentiles | Measured | LCRN‡ (baseline and 12 months follow-up) Longer term: CTRU accessing routine data | x | x | x | |
| Sibling age-adjusted and sex-adjusted BMI, height (m), weight (kg), unadjusted BMI and weight/BMI percentiles | | CTRU accessing routine data | | | x | |
| Parent self-efficacy | Dumka[27] | LCRN | x | x | | |
| Family eating/activities | Golan[28] | LCRN | x | x | | |
| Feeding questionnaire | Baughcum[29] | LCRN | x | x | | |
| Dental health (child) | Dental questionnaire | LCRN | x | x | | |
| Parent height and weight: | Measured | LCRN | x | x | | |
| Parent waist circumference | Measured | LCRN | x | x | | |
| Staff: | | | | | | |
| Staff screening | Staff screening form | Staff self-complete at centre | | | | |
| Staff height and weight | Measured | Self-measure | x | x | | |
| Staff waist circumference | Measured | Self-measure | X | x | | |
| Routine data: | | | | | | |
| NCMP Child and sibling data (trial participants) | | CTRU | | | x | |
| NCMP regional child data (not trial participants) | | CTRU | | | x | |

*Medium-term outcomes will be gathered from routinely collected data (from health visitors and/or the NCMP).
†Longer-term outcomes will be based on matched cohorts of (Millennium Cohort Study).[21]
‡Local Clinical Research Network (England only). Other devolved nations will train researchers/equivalent staff to collect data.
BMI, body mass index; CTRU, Clinical Trials Research Unit; LCRN, local clinical research network; NCMP, National Child Measurement Programme.

## Outcomes

The primary outcome for the trial is child age-adjusted and sex-adjusted BMI z-score assessed at 12 months post parent registration.

Secondary outcomes will be assessed at the same time point (defined short term), after 3 years (medium term) and will be forecasted across the life course (longer term). Individual-level outcomes are summarised in table 1 (see online supplemental table 1) for a full list of outcomes and measures).

## Data collection

Table 1 provides information about the types of data collected and how these are collected. Baseline and follow-up data (12 months) will be collected within participant homes by trained LCRN staff (or equivalent trained staff in the devolved nations). Where preferred by the participant, there will be options to collect data in the children's centre. The majority of questionnaires completed during home visits will be interviewer administered. Centre-level outcomes will be completed by a nominated member of centre staff (usually the centre manager or HENRY facilitator). Staff-level data will be self-completed and entered directly on to the online database.

## Sample size

41 centres per arm (from 10 to 14 local authorities), each recruiting 12 parents on average (4 parents from 3 programmes, 984 parents in total) will provide 90% power to detect a small standardised effect size of 0.27 as per previous trials (25–28) for BMI z-score at a 5% significance level, assuming an ICC of 0.03 (16, 25, 29–31) to

account for clustering by centre, a coefficient of variation of 0.48 to account for variation in centre recruitment and 20% loss to follow-up.[18]

## Withdrawal of consent

Centres and/or local authorities can withdraw at any point during the trial. Data collected up to the time of withdrawal will be retained for analysis and data from parents and staff will still be collected provided they have not withdrawn consent themselves. Centres may stop delivering the HENRY programme during the trial period independently from the trial (eg, centre closures, restructuring). Trial procedures will continue in this eventuality and all recruited parents and staff will remain in the trial (and data will continue to be collected from them) unless they actively withdraw. Where parents or staff wish to withdraw, there will be clarification whether this is withdrawal from short-term trial data collection, or from medium-term trial data collection (using routine data) or a combination of these. Non-attendance at HENRY intervention sessions are not classed as a withdrawal from the trial. All parents who withdraw from HENRY intervention or who do not attend the intervention will still be followed up for data collection unless they specifically express a wish to withdraw from trial processes.

## Data analysis plan

Analysis will be carried out on the intention-to-treat (ITT) population defined as all local authorities/centres randomised and all parents registered to the trial, regardless of adherence to the protocol, withdrawal of consent or losses to follow-up. A two-sided 5% significance level will be used for statistical endpoint comparisons. No interim analyses are planned, except for safety data that are required for review by the data monitoring and ethics committee.

The flow of local authorities, centres and participants through the trial will be presented in a Consolidated Standards of Reporting Trials diagram.

The primary outcome, child age-adjusted and sex-adjusted BMI (BMI z-score) will be analysed using a multilevel linear regression with children nested within centres and centres treated as a random effect. The model will be adjusted for the following fixed effects: centre-level stratification factors, important parent-level and child-level covariates (eg, baseline child BMI z-score and sex, parent BMI), and other relevant known predictors of outcome. Missing data will be imputed at the individual participant level where appropriate. Estimated mean differences will be reported with 95% confidence intervals, p values and ICCs. Model diagnostics will be used to check the underlying assumptions of the model and alternative methodology will be used if required.

Sensitivity analyses of the primary endpoint will be conducted to assess the impact of missing data, the choice of imputation model and the missing at random assumption, as appropriate. If contamination between intervention and control centres is identified, a sensitivity analysis excluding the relevant control centres will be conducted.

If numbers allow, exploratory subgroup analysis will examine differences in intervention effect between different socioeconomic and ethnic groups, and also between online and face-to-face delivery of HENRY.

For secondary outcomes, summary statistics will be presented at baseline and 12 months post parent registration overall and by arm (means, SD, medians, minimum, maximum and quartiles for continuous variables, and counts and percentages for categorical variables).

Medium-term analysis (3 years post parent recruitment) will compare regional population-level BMI z-score (trial local authority areas) with that of trial participants and siblings (HENRY and control) to investigate differences in BMI z-score. Analysis will use the same approach as the primary outcome for different outcome types, using multilevel regression with multiple imputation for missing data.

## Internal pilot

Descriptive analysis of the internal pilot against progression criteria will take place at three separate time points as described in table 2 and discussed with the trial steering committee (TSC) to inform a decision on the modification or continuation of the trial.

**Table 2** Internal pilot progression criteria

| Criteria | Green (go) | Amber (review) | Red (stop) |
|---|---|---|---|
| Centre recruitment: Centres open within 12 months of starting centre recruitment (including data up to the end of month 19). A centre will be defined as open if it has been randomised and is open to recruitment. | ≥54 | 42–53 | <42 |
| Parent recruitment: Average number of parents recruited per programme/equivalent (including data up to the end of month 23, allowing for 6 months of parent recruitment). | ≥4 | 3–<4 | <3 |
| HENRY programme delivery: Percentage of intervention centres having started delivery of at least one programme (including data up to the end of month 27, allowing for 18 months from starting centre recruitment). The denominator will include all centres allocated to the intervention arm. The numerator will be all of those intervention centres that have started to deliver at least one programme, defined as delivering at least one session to parents. | ≥80% | 50%–80% | <50% |

HENRY, Health, Exercise and Nutrition for the Really Young.

## Economic evaluation

Economic analysis will be conducted in three stages: (1) 12-month short-term within-trial cost-effectiveness analysis of the incremental cost per unit change in BMI z-score; (2) 3-year cost-effectiveness analysis of the incremental cost per unit change in BMI z-score with additional routinely collected NCMP data and (3) longer-term estimates of BMI z-score trajectories and healthcare use using a matched cohort of Millennium Cohort Study (MCS) participants.[21] The MCS dataset provides a unique and high-quality resource that enables us to project long-term BMI trajectories and healthcare resource use into early adulthood at the individual level.

The overarching aim of the economic analysis is to reduce decision uncertainty about whether or not HENRY should be commissioned. As we increase the time horizon of the analysis, it is anticipated that decision uncertainty will be reduced in the sense that it will become much clearer whether or not HENRY should be commissioned. Primary analyses will adopt an NHS and local authority perspective. Supplementary analyses will adopt a wider societal perspective by assessing household costs and productivity losses. Spill-over benefits to parents will also be captured by calculating differences between the two treatment groups in terms of quality-adjusted life-years over 12 months. Utility scores will be measured using both the EQ-5D-5L[22] and the ICECAP-A.[23] All analyses will be conducted using the ITT population. Seemingly unrelated regression will be used to account for the correlation between costs and outcome measures. Multilevel models will be used to account for children nested within centres. Decisions about which child-level, parent-level and centre-level covariates to include in the models will be made after assessing differences in baseline characteristics, and through discussion with the trial statisticians. Patterns (and reasons) of missing data will be investigated in collaboration with the trial statisticians and appropriate imputation techniques will be used. Supplementary analyses will assess whether there are differences in costs and effectiveness by the intervention delivery method by including an interaction term between method (ie, online vs face to face) and the treatment variable. In order to address whether or not the intervention is equally cost-effective among children living in the most deprived areas and households, analyses will also use interaction terms for household-level socioeconomic status and for area-level deprivation (≤10%/>10% ranking within Index of Multiple Deprivation at the lower super output area). To assess whether a parent/carer's propensity to commit to engage with HENRY was affected by their attitude to risk or perceptions about their long-term health prospects, analyses will also use interaction terms for self-reported measures of these factors based on questions used in the English Longitudinal Study of Ageing[24] and German Socioeconomic Panel.[25]

Uncertainty in our cost-effectiveness estimates will be characterised by presenting bootstrapped estimates on CEACs using a wide range of different cost-effectiveness thresholds (£/BMI change). While acknowledging that the true WTP threshold is unobserved, the choice of thresholds would be based on existing literature on the cost-effectiveness of comparable interventions and expert opinion. A version of the CEAC which shows the probability of the intervention being cost saving (ie, in the SE quadrant), regardless of the willingness-to-pay (WTP) threshold, will also be presented. One way sensitivity analyses will include an assessment of varying the intervention cost and of using alternative methods and approaches to matching the trial data with participants in the MCS dataset. Results for all economic analyses will be reported in accordance with the Consolidated Health Economic Evaluation Reporting Standards guidelines[26] using Stata (StataCorp, V.18.0) and/or R (R V.4.3).

## Systems-based process evaluation

HENRY is a complex intervention that can be viewed as an event within a system.[27] Therefore, we will adopt a complex systems perspective, using a framework for qualitative systems process evaluations[28] and quantitative systems evaluation, embedded within a mixed methods process evaluation.[29] The evaluation will consist of the two-stage qualitative systems approach: (1) initial systems mapping and (2) analysis of the system within which HENRY is embedded. It will also include a traditional process evaluation to understand context, mechanisms and implementation of HENRY[29] and quantitative systems modelling.

The systems map will be constructed using data collected at a stakeholder workshop (n=30–40) at the start of the RCT in which we will identify: structure (eg, levels of national, regional and local); elements (eg, national public health priorities around childhood health, obesity and parenting; local authority priorities and funding situation; health visitor responsibilities; organisations providing HENRY; welfare benefits systems; historical events affecting childhood obesity); relationships and interactions between elements; and boundaries (what is inside and outside the system). Invited stakeholders will include (but will not be restricted to) childcare practitioners, public health specialists, commissioners, parents, health visitors and local and national early years organisations/delivery teams. The overall map may ultimately consist of a number of submaps that will highlight causal inter-relationships between each of the elements of these areas. It may also present HENRY at different levels (the local authority, centre, parent/child and others) identified via this research. These maps will comprise fundamental systems thinking 'building blocks' in the form of causal-loop diagrams and/or stock-and-flow structures as appropriate. Each set of diagrams (or system maps) will seek to relate parameter/variable components of the local system through positive/negative causality and ordinalities such that a dynamic representation of inter-relationships can be viewed graphically.

The second stage of qualitative systems process evaluation will explore how the system and its elements change

over time in response to HENRY. Six months prior to the end of the RCT, we will undertake qualitative telephone interviews/virtual interviews with 20–24 national, regional and local key stakeholders representing elements of the stage 1 map to discuss the system structure and elements, how they have changed over time, the way the system responded to HENRY, and how responses amplified or dampened HENRY's impacts. We will also explore how HENRY affected or was affected by potential strategies for addressing health inequalities.

To complement McGill's qualitative systems process evaluation, we will also focus on understanding the reach of HENRY within the target population, how HENRY works, potential contamination and how it has been implemented within the RCT.[29] We will undertake semi-structured interviews with staff providing HENRY (n=12–16) to explore their views of the feasibility of HENRY, how it has been delivered in different centres over time (variation in implementation), access to HENRY for different socioeconomic and ethnic minority groups (reach), perceptions of differential health outcomes for different socioeconomic and ethnic minority groups (ie, how it addresses health inequalities or not), and views of potential contamination in the RCT. We will also undertake semistructured interviews with parents (n=20–24) attending HENRY to explore acceptability of the intervention and perceptions of how it facilitated health improvement or not. In both sets of interviews, we will explore mechanisms of impact (how HENRY has affected parenting, nutrition and weight of children), facilitators and barriers to delivering or attending HENRY, reach in terms of accessing those most in need and implementation in practice. We will also include a systems lens by putting context at the centre of the interviews and explore the impact of system elements on HENRY, specifically asking about how elements of the system documented in stage 1 interacted with HENRY. These interviews may affect our understanding of why HENRY was implemented in different ways in different localities or at different times, and the extent to which health inequalities have been addressed by HENRY.

Quantitative process data will be gathered to summarise attendance and drop out by centre and overall and will support the monitoring and evaluation of contamination. Thus, in addition to exploring contamination through qualitative interviews, data on staff movement and sharing of HENRY related messages will be captured. Key to this will be an online HENRY-trained staff survey at two time periods (towards the end of the internal pilot and at the end of the RCT intervention period) asking whether, how and when staff shared HENRY practice outside their centre.

Qualitative data will be analysed prior to the trial outcome analysis using the framework approach. An initial thematic framework will be based on familiarisation of a range of transcripts, analysis of qualitative interviews from the feasibility study,[30] and the process evaluation framework (mechanisms, context and implementation).[29] Quantitative evaluation will combine trial and process data to apply a causality-based model of obesity prevention to develop a systems archetype.[31] Analysis will incorporate Morphological Analysis and Fuzzy Cognitive Mapping techniques[32] to evaluate and assign inter-relationships between components through causal weights and network directionality. We will examine the range of operating conditions of the HENRY system model to classify its overall dynamic behaviour (identifying which Systems Archetype HENRY most closely resembles[31]; providing a basis for comparison to other obesity models (eg, Foresight) to explain how HENRY may disrupt obesity prevalence in childhood).

## Patient and public involvement

We will consult closely with a diverse group of parents with young children from a local children's centre who supported the design and research processes for our funding application. Activities will include workshop-style meetings timed to project milestones and in line with TSC meetings (approximately twice per year of the trial). Each workshop will have a specific aim and associated set of activities including brainstorming and group discussion. We will share workshop outcomes with research team members, including the steering committee, and, where applicable, incorporate them in research activities. To begin, the group will be asked to consider their needs for involvement, including training or practical requirements. The dedicated public involvement lead (WB) will be responsible for setting and overseeing the overall strategy. Proposed activities will be embedded in all stages of the research. For example, during recruitment, we will discuss unanticipated barriers (eg, COVID-19-related issues) so that we can help mitigate these. The group will provide input into the process evaluation work; for example, ensuring concepts and language used during systems mapping workshops is appropriate. They will also be involved in interpreting trial results in a way most relevant to parents and help to design and disseminate trial outputs. Parents will be remunerated for their time using NIHR guidelines.[33]

## Ethics and dissemination

The trial will adhere to ethical principles, approach, aims and methods of the ESRC research framework. Ethical approval was obtained by the University of York, Health Sciences' Research Governance Committee (HSRG-C/2022/537/E). HRA approval was also obtained to allow NHS (eg, children's centres sitting within integrated care organisations) sites to take part in the trial (Integrated Research Application System (ID: 317992). Serious adverse events are not anticipated; however, we have convened an independent TSC and separate data monitoring and ethics committee (DMEC) to monitor trial progress and adherence (TSC) and ethical issues and safety (DMEC).

Our dissemination strategy includes both the production of regular progress reports and outputs to share with

local and national governments, and final reports and publications. For example, we will disseminate findings of the systems work via meetings and conferences, and via publications (ie, not restricted to the end of trial). Our strategy also includes regular public-facing newsletters, social media outputs and attendance at festival events. Final outputs will be shared via a dissemination event, policy briefings and academic publications/presentations.

**Author affiliations**
[1]Hull York Medical School, University of York, York, UK
[2]Department of Health Sciences, University of York, York, UK
[3]Clinical Trials Research Unit, Leeds Institute for Clinical Trials Research, University of Leeds, Leeds, UK
[4]Academic Unit of Health Economics Leeds Institute of Health Sciences, University of Leeds, Leeds, UK
[5]Sheffield centre for Health and Related Research, University of Sheffield, Sheffield, UK
[6]Leeds Institute for Health Sciences, University of Leeds, Leeds, UK
[7]School for Business and Society, University of York, York, UK

**Acknowledgements** We thank the members of the TSC, including Professor Peymane Adab (TSC Chair, University of Birmingham), Professor Amelia Lake (Teeside University), Professor Laura Cornelsen (London School of Hygiene and Tropical Medicine), Dr Nasima Akhter (Teeside University) and Lina Alrefaai (parent representative), Paul Atkin (CRN Yorkshire), Professor Simon Coulton (University of Kent), Lauren Surtees (parent representative). We are also grateful to our DMEC members, Dr Sarah Seaton (University of Leicester), Professor Miranda Pallan (University of Birmingham) and Dr Sarah Crozier (University of Southampton).

**Contributors** MB: conceptualisation, funding acquisition, oversight (study lead), methodology, writing–original draft, writing–reviewing and editing; WB: conceptualisation, funding acquisition, project management, PPI lead, writing–original draft, writing–reviewing and editing; MC: funding acquisition, methodology (statistical lead), writing–original draft, writing–reviewing and editing; AM: funding acquisition, methodology (economics lead), writing–original draft, writing–reviewing and editing; BC: funding acquisition, methodology (analysis), writing–original draft, writing–reviewing and editing; DG-W: methodology (trial management), writing–original draft, writing–reviewing and editing; AF: methodology (systems process evaluation), writing–original draft, writing–reviewing and editing; TAW: methodology (trial operations), writing–original draft, writing–reviewing and editing; PG: methodology (systems quantitative evaluation lead), writing–reviewing and editing; AO'C: funding acquisition, methodology (systems process evaluation lead), writing–original draft, writing–reviewing and editing.

**Funding** This study/project is funded by the NIHR [name of NIHR programme (NIHR135081).

**Disclaimer** The views expressed are those of the authors and not necessarily those of the NIHR or the Department of Health and Social Care.

**Competing interests** None declared.

**Patient and public involvement** Patients and/or the public were involved in the design, or conduct, or reporting, or dissemination plans of this research. Refer to the Methods section for further details.

**Patient consent for publication** Not applicable.

**Provenance and peer review** Not commissioned; externally peer reviewed.

**ORCID iDs**
Maria Bryant http://orcid.org/0000-0001-7690-4098
Bethan Copsey http://orcid.org/0000-0001-9783-6549
Dawn Groves-Williams http://orcid.org/0000-0002-4834-8503
Alexis Foster http://orcid.org/0000-0002-7978-2791
Alicia O'Cathain http://orcid.org/0000-0003-4033-506X

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
