## [Reviewer comments · BMJ Open]

ARTICLE DETAILS

TITLE (PROVISIONAL)	Effectiveness and cost-effectiveness of a sustainable obesity prevention programme for pre-school children delivered at scale 'HENRY' (Health, Exercise, Nutrition for the Really Young): Protocol for the HENRY III Cluster Randomised Controlled Trial
AUTHORS	Bryant, Maria; Burton, Wendy; Collinson, Michelle; Martin, Adam; Copsey, Bethan; Groves-Williams, Dawn; Foster, Alexis; Willis, Thomas A; Garnett, Philip; O'Cathain, Alicia

VERSION 1 – REVIEW

REVIEWER	Malden, Stephen The University of Edinburgh School of Health in Social Science, Scottish Collaboration for Public Health Research and Policy
REVIEW RETURNED	28-Nov-2023

GENERAL COMMENTS	Thank you for the opportunity to review this manuscript, which describes a protocol for the HENRY pre-school obesity prevention cluster RCT. The manuscript is well written, and the study appears to be well designed. Only a couple of minor questions/suggestions: - I may have missed it, but it appears you will only be including Local Authority nurseries in the trial (i.e. not private nurseries) is this correct? If so, this should be justified - especially given the systems-based approach that is being adopted, as this will be a large portion of the "system" not included. Process evaluation: This appears very comprehensive. However, it would be helpful to know in what order process and outcome data are to be analysed. Ideally, process data should be analysed prior to outcome data to avoid potential biases in interpretation of findings (see MRC guidelines on process evaluation of complex interventions). Thank you.
---

REVIEWER	Dobell, Alexandra University of Birmingham, Applied Health Research
REVIEW RETURNED	29-Dec-2023

GENERAL COMMENTS	I would like to thank the authors for the opportunity to review 'Evaluation of a sustainable obesity prevention programme for pre-school children delivered at scale 'HENRY' (Health, Exercise,
---

	Nutrition for the Really Young): Protocol for the HENRY III Randomised Controlled Trial'. The protocol provides key information about the intended evaluation of the effectiveness and cost-effectiveness of the HENRY III programme, in addition to the systems process evaluation. This research intends to provide vital information about a long-standing intervention and evidence for its effectiveness at reducing obesity, especially in deprived populations. The full protocol provided with the manuscript details procedures for the research in detail. I personally believe the manuscript needs very minor editing, and some comments the authors may wish to address are included below. Abstract - Introduction Change: 'A fifth of children start school with already living with' to 'A fifth of children start school already living with' 'One of these (HENRY)' to 'One of these programmes (HENRY)' Add a comma after 'since 2008' General note: HENRY (in methods and analysis) becomes HENRY III in strengths and limitations- ensure consistency Introduction 'This is the largest annual increase in the UK since the National Child Measurement Programme (NCMP) recording began'. - The sentence before states pre and post COVID, however, this COVID was spread across more than one year, I feel this sentence needs to state years instead of pre and post covid. '(including mental health services and those used to treat and manage respiratory diseases such as asthma' -remove (and replace with comma. 'Given that obesity is caused by a wide range of factors, this is unsurprising and means that the role and cost-effectiveness of locally delivered programmes within a large and complex socio-economic and public health system is uncertain'- I think here it would be good to list some of the broad factors that contribute to obesity. May be worth thinking about replacing PHE as it ceases to exist?- it has been replaced by UK Health Security Agency and Office for Health Improvement and Disparities. Or change wording to Public health England ADVOCATED? 'Public Health England (PHE) advocates systems approaches; encouraging local areas to adopt a range of interconnecting interventions and policies inside and outside the healthcare sector to collectively tackle obesity and related health inequalities' - I feel this sentence needs a citation. Ensure the introduction is consistently discussing childhood obesity- collective obesity is discussed in the second paragraph before introducing the programme in the third paragraph, focussing on children again. I think more information about the history of the HENRY programme and how it as developed would be good context for readers. More information about obesity systems should be included in this introduction, so the reader can understand what the HENRY programme is aiming to effect. I feel it is assumed that readers will already understand the system. This does not need to be detailed but broader information should be included, as it is discussed regularly. This will also lead nicely into the aims and objectives as they state parts of the system that are being measured in the evaluation. Methods and analysis
--	---

	What kinds of 'other associations' do you expect may be recruited? How will you ensure a spread of LA's across the UK? Will centres receive any money for their participation in the trial? 'plan to run 3 programmes during the trial'- how long will the trial last, I don't think this has been clear in the lead up to this point? 'Centres that are either currently delivering HENRY or have previously delivered HENRY in the last two years are not eligible – even if they have never delivered HENRY in those centres.' This sentence doesn't make sense, please revisit. Data collection 'Baseline and follow-up data (12 months) will be collected within participant homes by trained Local Clinical Research Network (LCRN) staff'- can data collection also happen at the centres? Public Involvement How will these parents be recruited and how many meetings do you anticipate holding with them. Will they receive remuneration for the time?
--	---

REVIEWER	Sinha, Madhumita NIH Phoenix
REVIEW RETURNED	02-Jan-2024

GENERAL COMMENTS	Title of Study Protocol: Evaluation of a sustainable obesity prevention programme for pre-school children delivered at scale 'HENRY' (Health, Exercise, Nutrition for the Really Young): Protocol for the HENRY III Randomised Controlled Trial SUMMARY: The current protocol aims to study the clinical and economic impact of a publicly funded healthy lifestyle/behavioral intervention program (HENRY) for parents and caregivers of preschoolers (6 month to 5 years of age) in socioeconomically disadvantaged communities in the United Kingdom (UK). This program has been delivered for the past several years and follows a community based participatory model. The proposed multi-center randomized controlled trial would enable the investigators to conduct additional systems-based process evaluation in a prospective manner and conduct a cost effectiveness analysis to assess the future viability of the HENRY program from an economic perspective and likely inform healthcare policy makers, This is an important study from a public health perspective. OVERALL COMMENTS. The current study follows a prior feasibility study (Ref 1. This study has not been cited in the Bibliography) that showed the study team was able to engage the local authorities (stakeholders), parents and children centers successfully in study participation, and the study had a robust data collection protocol. However, there was a selection bias that impacted the primary outcome in this feasibility study with children in the control group having a lower BMI z-score than those in the intervention arm at baseline. The authors noted that the trend in mean BMI z-score reduction towards a healthy level among the intervention group pointed in the anticipated direction. Although I agree with the cluster randomization model adopted in this study, one way to mitigate this selection bias could also be to assign families at random to the intervention and control groups in a 1:1 ratio, stratified by clinical sites (children's center). Randomization within each site could be stratified by BMI z-score categories (such as normal, overweight, obese categories by z-scores or percentiles), of the reference or index child. If two
--

siblings from the same family are participants and fit the age category for HENRY, one could use the highest BMI z-score of the two siblings for stratification. Randomization would not be revealed to staff or participants until completion of baseline measurements. This method is likely to create more balanced groups and the primary outcome of change in BMI z-score at 12-months and 3-years following intervention can be compared accurately. The authors commented that "If this selection bias was repeated in a future definitive trial, the primary outcome of 'difference between BMI z-score at follow-up' may not show the true effect of the HENRY programme". (Ref 1). In the proposed study I do not see this issue being addressed specifically. In the randomization plan (page 8, last para) the characteristics to be considered in creating balanced groups are noted to be: size of the center, area level of ethnicity and deprivation. Since the intervention is brief and delivered over 8 x 2-hour sessions without additional booster sessions, and the primary outcome of change in BMI z-score will be assessed at 12 and 36 months post parent registration, documenting a significant change in BMI z-score from baseline to 12-months and at 36-months (knowledge retention over an at least 2 year period with no additional reinforcement) seems to be an outcome that is too ambitious and difficult to achieve.

Furthermore, since the current RCT will provide data for a robust economic analysis (short term, medium and long term) and will estimate a 12-month within trial calculation of cost per unit change in BMI z-score, it may be premature to base the economic analysis on a target that may not be achievable. The strength of this study being the economic analysis and systems process assessment.

SECTION SPECIFIC COMMENTS

1. ABSTRACT:

The introduction section needs rewriting with attention to framing of sentences that have some grammatical/sentence structuring errors. Some phrases like "early year obesity system" is unfamiliar to this reviewer.

2. INTRODUCTION

The authors have emphasized the need for community-based programs in obesity prevention since chronic conditions such as childhood obesity have to be addressed from an eco-bio-developmental perspective and the child who is the target of change does not thrive in isolation but within microsystems that comprise their family, school, and larger ecosystems of community and society as a whole (Ref 2). However, community involvement in this proposed RCT is not clearly articulated especially in what seems to be a diverse study population. The only additional stakeholders who can participate in this study seem to be the clinic staff, and parents who will be part of the study and parent advisory group will support "delivery and dissemination". The authors do mention that the parent HENRY program did have input from governmental partners and public. However, additional stakeholder involvement/inputs such as local community leaders, schoolteachers etc. in program assessment, improvement, and future dissemination is not elucidated.

In the last para (page 5) of introduction, the authors mention that in their current evaluation, child, and sibling ?obesity outcomes will be assessed in the short and medium term from anthropometric data collected in the study and long-term using modelling data

	compared with secondary datasets. It seems from reading the methodology, that the unit of assessment here is the family (index child, sibling, parent's adiposity measures) so this has to be clarified whether the outcome is only assessed in the index/reference child, reference child and sibling/s or the family as a unit since in this very young pediatric cohort, the educational intervention including motivational interviewing is targeted towards the parents or caregivers to presumably to change the health behavior of the entire family (as enumerated in the secondary outcome list). Aims and objectives: The assessment of change in BMI z-scores of the reference or index child at 3-years, and of siblings (BMI z-scores) and parents (BMI) at 12-months and 3-years should be mentioned as part of the secondary objectives if that is the case and the primary objective is to only assess the BMI z-score change in the reference child at 12-months post parents registration. Methods and analysis: The eligibility criteria are adequately outlined. Please see my previous comments on randomization to prevent a selection bias. The availability of interpreters or relatives for non-English speaking participants is articulated, however from prior experience this may be a significant barrier particularly during motivational interviewing sessions (?group ?individual). Active intervention: Although I understand that the HENRY program has been developed and deployed over the past several years in the UK, the reader who is unfamiliar with the current program may benefit from additional information: since the population is mixed. Has this program been tailored to specific racial/ethnic minority groups where dietary practices may be diverse? Since HENRY is a multicomponent program what are the exact elements of this behavioral intervention program? (?motivational interviewing for individual parent/child units, ?supervised group education classes covering a behavioral modification approach toward improving nutrition practices and physical activities'). Also, a Figure specifying intervention and follow-up timeline would render clarity to how the program will be deployed, and assessments done. For example a parent may register for the program in January 2024, but the actual intervention at that center may not start until another 6 months, if the intervention is over lets say 2-months this child may not have enough time to record an improvement in their BMI compared to another child where the center is well functioning and staff training, intervention occurs promptly and there is enough time to document a treatment effect. A figure with timelines would be helpful. Also, it seems that the intervention is complete after the 8 2-hour sessions, most lifestyle programs have additional booster sessions that improves health education information retention. Data collection: The clinical measures to assess adiposity in children/parents/staff would be weight, height, and waist circumference. Would additional clinical measures to assess general metabolic health such as blood pressure, presence of clinical signs of insulin resistance such as acanthosis in older children be collected? Are there any plans for DEXA scans in a subset of participants? It seems that no additional biochemical tests are planned in this study. But clinical history such as history of gestational diabetes in
--	--

	the mother, birth weight, breast feeding and others that are established predictors for excess adiposity especially in early life may be additional important data points for this study. Patient recruitment and retention. From reviewing the feasibility study (Ref 1) it is apparent that the study faced barriers with participant recruitment strategies and did not hit their target. The current protocol does not elaborate on recruitment and more importantly retention strategies in this underserved cohort who suffer from many barriers to accessing healthcare. COST ANALYSIS I commend the authors for their intention to conduct an economic evaluation of an established publicly funded health intervention program that would inform policy makers if the program is cost effective. The authors should provide more details about the methodology of the proposed cost-effectiveness analysis with respect to how the cost-effectiveness acceptability curves will be developed and at what level of willingness to pay (WTP). What formal methods of uncertainty analysis using sensitivity/threshold analyses will be done? Will Monte Carlo analysis be done using different types of distribution? Also, the type of software that will be used should be mentioned. Ref 1. Bryant, M., Collinson, M., Burton, W. et al. Cluster randomised controlled feasibility study of HENRY: a community-based intervention aimed at reducing obesity rates in preschool children. Pilot Feasibility Stud 7, 59 (2021). https://doi.org/10.1186/s40814-021-00798-z Ref 2. Davison KK, Birch LL. Childhood overweight: a contextual model and recommendations for future research. Obes Rev 2001;2: 159–171.
--	--

VERSION 1 – AUTHOR RESPONSE

Reviewer: 1

Dr. Stephen Malden, The University of Edinburgh School of Health in Social Science
Comments to the Author:

Thank you for the opportunity to review this manuscript, which describes a protocol for the HENRY pre-school obesity prevention cluster RCT.

The manuscript is well written, and the study appears to be well designed.

Thank you

Only a couple of minor questions/suggestions:

- I may have missed it, but it appears you will only be including Local Authority nurseries in the trial (i.e. not private nurseries) is this correct? If so, this should be justified - especially given the systems-based approach that is being adopted, as this will be a large portion of the "system" not included.

We have clarified that either type of nursery is eligible within Section 'Setting and recruitment/ Eligibility Criteria/ Centres'.

Process evaluation: This appears very comprehensive. However, it would be helpful to know in what order process and outcome data are to be analysed. Ideally, process data should be analysed prior to outcome data to avoid potential biases in interpretation of findings (see MRC guidelines on process evaluation of complex interventions).

We are able to confirm that the process evaluation analysis will occur prior to the analysis of the main trial. This has been clarified within Section 'Systems-based process evaluation'.

Reviewer: 2

Dr. Alexandra Dobell, University of Birmingham

Comments to the Author:

BMJ Review

I would like to thank the authors for the opportunity to review 'Evaluation of a sustainable obesity prevention programme for pre-school children delivered at scale 'HENRY' (Health, Exercise, Nutrition for the Really Young): Protocol for the HENRY III Randomised Controlled Trial'. The protocol provides key information about the intended evaluation of the effectiveness and cost-effectiveness of the HENRY III programme, in addition to the systems process evaluation. This research intends to provide vital information about a long-standing intervention and evidence for its effectiveness at reducing obesity, especially in deprived populations. The full protocol provided with the manuscript details procedures for the research in detail.

I personally believe the manuscript needs very minor editing, and some comments the authors may wish to address are included below.

Abstract - Introduction

Change: 'A fifth of children start school with already living with' to 'A fifth of children start school already living with'

Amended

'One of these (HENRY)' to 'One of these programmes (HENRY)'

Amended

Add a comma after 'since 2008'

We feel that the 'and' negates the need for a comma in this instance

General note: HENRY (in methods and analysis) becomes HENRY III in strengths and limitations- ensure consistency

We have used HENRY III to denote the trial name and HENRY when describing the intervention.

Introduction

'This is the largest annual increase in the UK since the National Child Measurement

Programme (NCMP) recording began'. - The sentence before states pre and post COVID, however, this COVID was spread across more than one year, I feel this sentence needs to state years instead of pre and post covid.

Amended

'(including mental health services and those used to treat and manage respiratory diseases such as asthma' -remove (and replace with comma.

Amended

'Given that obesity is caused by a wide range of factors, this is unsurprising and means that the role and cost-effectiveness of locally delivered programmes within a large and complex socio-economic and public health system is uncertain'- I think here it would be good to list some of the broad factors that contribute to obesity.

Added

May be worth thinking about replacing PHE as it ceases to exist?- it has been replaced by UK Health Security Agency and Office for Health Improvement and Disparities. Or change wording to Public health England ADVOCATED?

As there is also uncertainty with longevity of OHID, we have amended to 'advocated' as suggested.

'Public Health England (PHE) advocates systems approaches; encouraging local areas to adopt a range of interconnecting interventions and policies inside and outside the healthcare sector to collectively tackle obesity and related health inequalities' - I feel this sentence needs a citation.

Added – now citation (11).

Ensure the introduction is consistently discussing childhood obesity- collective obesity is discussed in the second paragraph before introducing the programme in the third paragraph, focussing on children again.

Amended in second paragraph and added for consistency to start of third paragraph.

I think more information about the history of the HENRY programme and how it as developed would be good context for readers.

As we are keen to stay concise, we feel that the level of information is sufficient here (with corresponding references) given that we dedicate space to discussing the intervention in the Methods section (Active intervention).

More information about obesity systems should be included in this introduction, so the reader can understand what the HENRY programme is aiming to effect. I feel it is assumed that readers will already understand the system. This does not need to be detailed but broader information should be included, as it is discussed regularly. This will also lead nicely into the aims and objectives as they state parts of the system that are being measured in the evaluation.

We have added a little more text on obesity systems (which we feel is also covered in providing concrete examples of factors that influence obesity) in the 2nd paragraph of the introduction. However, we have kept this relatively broad to ensure that our introduction is concise.

Methods and analysis

What kinds of ‘other associations’ do you expect may be recruited?

Added examples in Section ‘Setting and recruitment/Local authority and centre recruitment’.

How will you ensure a spread of LA’s across the UK?

We have clarified the process for recruiting local areas within Section ‘Setting and recruitment/Local authority and centre recruitment’. We are not able to ensure a spread of LAs across the UK, as our method of recruitment is dependant upon the needs and resources of LAs. In other words, provided an area meets the eligibility criteria, we will consider them for recruitment.

Will centres receive any money for their participation in the trial?

Participant screening and recruitment will be remunerated based on NIHR CRN guidance for service support costs. This information is now included, with a link to an appropriate citation.

‘plan to run 3 programmes during the trial’- how long will the trial last, I don’t think this has been clear in the lead up to this point?

Clarified in the Section ‘Eligibility criteria’

‘Centres that are either currently delivering HENRY or have previously delivered HENRY in the last two years are not eligible – even if they have never delivered HENRY in those centres.’ This sentence doesn’t make sense, please revisit.

Agree this is confusing and have therefore amended this sentence.

Data collection

'Baseline and follow-up data (12 months) will be collected within participant homes by trained Local Clinical Research Network (LCRN) staff'- can data collection also happen at the centres?

Yes, at the request of the parent. This has been added to Section 'Data collection' though, based on our feasibility findings, we do not feel that this will happen extensively.

Public Involvement

How will these parents be recruited and how many meetings do you anticipate holding with them. Will they receive remuneration for the time?

We have added clarification of all of these points in the Section 'Patient and Public involvement, including a citation (33) regarding guidance of involvement and payment.

Reviewer: 3

Dr. Madhumita Sinha, NIH Phoenix

Comments to the Author:

Title of Study Protocol: Evaluation of a sustainable obesity prevention programme for pre-school children delivered at scale 'HENRY' (Health, Exercise, Nutrition for the Really Young): Protocol for the HENRY III Randomised Controlled Trial

SUMMARY: The current protocol aims to study the clinical and economic impact of a publicly funded healthy lifestyle/behavioral intervention program (HENRY) for parents and caregivers of preschoolers (6 month to 5 years of age) in socioeconomically disadvantaged communities in the United Kingdom (UK). This program has been delivered for the past several years and follows a community based participatory model. The proposed multi-center randomized controlled trial would enable the investigators to conduct additional systems-based process evaluation in a prospective manner and conduct a cost effectiveness analysis to assess the future viability of the HENRY program from an economic perspective and likely inform healthcare policy makers, This is an important study from a public health perspective.

OVERALL COMMENTS. The current study follows a prior feasibility study (Ref 1. This study has not been cited in the Bibliography) that showed the study team was able to engage the local authorities (stakeholders), parents and children centers successfully in study participation, and the study had a robust data collection protocol.

This citation is reference number 18.

Although I agree with the cluster randomization model adopted in this study, one way to mitigate this selection bias could also be to assign families at random to the intervention and control groups in a 1:1 ratio, stratified by clinical sites (children's center). Randomization within each site could be stratified by BMI z-score categories (such as normal, overweight, obese categories by z-scores or percentiles), of the reference or index child. If two siblings from the same family are participants and fit the age category for HENRY, one could use the highest BMI z-score of the two siblings for stratification. Randomization would not be revealed to staff or participants until completion of baseline measurements. This method is likely to create more balanced groups and the primary outcome of change in BMI z-score at 12-months and 3-years following intervention can be compared accurately. The authors commented that "If this selection bias was repeated in a future definitive trial, the primary outcome of

‘difference between BMI z-score at follow-up’ may not show the true effect of the HENRY programme”. (Ref 1). In the proposed study I do not see this issue being addressed specifically. In the randomization plan (page 8, last para) the characteristics to be considered in creating balanced groups are noted to be: size of the center, area level of ethnicity and deprivation.

We agree that selection bias is a risk of this research. As described in our cover letter, the study team worked with the funders and Trial Steering Committee (TSC) members at length to consider the risk of selection bias and proposed a comprehensive mitigation plan. This involves regularly monitoring recruitment and participant characteristics throughout the trial in our Trial Management Group and TSC meetings, with strategies to deploy if any imbalance occurs. In addition to regular monitoring, the TSC will be provided with a comparison of the baseline characteristics (including baseline child BMI z-score, parent socio-economic status and ethnicity) to allow assessment of the level of selection bias which can be used to inform the decision to continue with the trial.

In terms of ensuring balance, we have now provided an account of the strategies we will use as a supplementary file.

Although the reviewer notes their agreement with the cluster design, their subsequent suggestion of randomising families within centres to intervention and control groups is an example of an individually randomised design which is not feasible in this setting. Interventions delivered to groups of parents risk contamination due to the way that parents offer one another support. Further, practitioners within centres who are trained in the HENRY approach will have interaction with many parents on a daily basis (including those assigned to controls if individually randomised). As the training supports them to incorporate HENRY into their everyday discussions with parents, it would not be feasible for them to share the same space with control parents without inadvertently sharing HENRY messages.

These factors (and the fact that it is a pragmatic evaluation of an intervention being implemented (and funded by) local government) mean that a cluster design was really our only feasible option. As a result, we are not able to stratify by individual factors. Instead, we have stratified by aggregated factors including area level deprivation. Our analysis model will adjust for parent and child level covariates (e.g. baseline child BMI z-score and sex, parent BMI).

Since the intervention is brief and delivered over 8 x 2-hour sessions without additional booster sessions, and the primary outcome of change in BMI z-score will be assessed at 12 and 36 months post parent registration, documenting a significant change in BMI z-score from baseline to 12-months and at 36-months (knowledge retention over an at least 2 year period with no additional reinforcement) seems to be an outcome that is too ambitious and difficult to achieve. Furthermore, since the current RCT will provide data for a robust economic analysis (short term, medium and long term) and will estimate a 12-month within trial calculation of cost per unit change in BMI z-score, it may be premature to base the economic analysis on a target that may not be achievable. The strength of this study being the economic analysis and systems process assessment.

This is an important point which we considered at the trial design stage, including discussion with the funding panel (at feasibility application). We concluded that a 12mth followup would be optimal because the intervention is designed to bring about changes in behaviour that are sustained for many years in real world situations, rather than delivering only a short-term treatment effect (i.e. more akin to effectiveness than efficacy). Due to the nature of the pathways through which the behaviour change leads to changes in weight, we also do not anticipate that the full potential of the intervention in terms of reduced BMI would fully materialise during the short intervention period.

Furthermore, from the perspective of the economic evaluation, it is our expectation that a health effect would need to be observed well past the end of the intervention period for there to be any chance of the intervention being good value for money. In the absence of any observable effect at 12 months, it is not envisaged that the intervention would be cost-effective since there would be no prospect of reductions in the likelihood of obesity-related chronic health conditions and healthcare expenditure during later childhood and early adulthood.

SECTION SPECIFIC COMMENTS

1. ABSTRACT:

The introduction section needs rewriting with attention to framing of sentences that have some grammatical/sentence structuring errors. Some phrases like “early year obesity system” is unfamiliar to this reviewer.

Amended as per Reviewer 1

2. INTRODUCTION

The authors have emphasized the need for community-based programs in obesity prevention since chronic conditions such as childhood obesity have to be addressed from an eco-bio-developmental perspective and the child who is the target of change does not thrive in isolation but within microsystems that comprise their family, school, and larger ecosystems of community and society as a whole (Ref 2). However, community involvement in this proposed RCT is not clearly articulated especially in what seems to be a diverse study population. The only additional stakeholders who can participate in this study seem to be the clinic staff, and parents who will be part of the study and parent advisory group will support “delivery and dissemination”. The authors do mention that the parent HENRY program did have input from governmental partners and public. However, additional stakeholder involvement/inputs such as local community leaders, schoolteachers etc. in program assessment, improvement, and future dissemination is not elucidated.

The relationships between children, parents, practitioners, centres, commissioners, health care professionals and other wider organisations is not currently known with regards to early years obesity systems. This is the focus of our planned systems based process evaluation where we plan to conduct a systems workshop with multiple and varied stakeholders at the start of the study and interviews with stakeholders near the end of the RCT. This is detailed in Section ‘Systems based process evaluation’ and we have added more detail to clarify the type of stakeholders. In this section, our existing text highlights the potential levels of influence that we will explore (e.g. *“levels of national, regional and local”*); *elements (e.g. national public health priorities around childhood health, obesity and parenting; local authority priorities and funding situation; health visitor responsibilities; organisations providing HENRY; welfare benefits systems; historical events affecting childhood*

obesity); relationships and interactions between elements; and boundaries (what is inside and outside the system”).

We have also added a brief summary to clarify the various factors that form part of the obesity system in our introduction which we feel helps to justify our systems work. We do not intend to influence these relationships, but to gain a better understanding of the system so that we can explore what role a programme like HENRY has in disrupting the system.

In the last para (page 5) of introduction, the authors mention that in their current evaluation, child, and sibling ?obesity outcomes will be assessed in the short and medium term from anthropometric data collected in the study and long-term using modelling data compared with secondary datasets. It seems from reading the methodology, that the unit of assessment here is the family (index child, sibling, parent’s adiposity measures) so this has to be clarified whether the outcome is only assessed in the index/reference child, reference child and sibling/s or the family as a unit since in this very young pediatric cohort, the educational intervention including motivational interviewing is targeted towards the parents or caregivers to presumably to change the health behavior of the entire family (as enumerated in the secondary outcome list).

Our primary unit of assessment is at the child level. We agree that this is confused with the addition of ‘sibling’ in the last page of the introduction and have amended this text. Our secondary analyses will include outcomes measured at multiple levels, including siblings, parents and HENRY practitioners. Information on these is detailed in the methods and in Table 1 (including the timeline).

Aims and objectives: The assessment of change in BMI z-scores of the reference or index child at 3-years, and of siblings (BMI z-scores) and parents (BMI) at 12-months and 3-years should be mentioned as part of the secondary objectives if that is the case and the primary objective is to only assess the BMI z-score change in the reference child at 12-months post parents registration.

Our primary aim is stated in the original text, to focus on child BMI at age 12 months. Parental, sibling and staff measures are included in the secondary objectives. Except for the primary objective, we have not included timelines, to prevent confusion given that there are multiple timelines. These are detailed elsewhere in the data collection section of the protocol.

Methods and analysis: The eligibility criteria are adequately outlined. Please see my previous comments on randomization to prevent a selection bias. The availability of interpreters or relatives for non-English speaking participants is articulated, however from prior experience this may be a significant barrier particularly during motivational interviewing sessions (?group ?individual).

We agree and ensure that we actively seek opportunities to involve a diverse range of participants.

Motivational interviewing can be included in the parenting group sessions as part of the HENRY approach. This pragmatic, independent evaluation is not able to influence intervention delivery beyond those defined in our eligibility criteria and randomisation. Intervention delivery teams are set up to serve their local populations and we will explore representation (as described) in our process evaluation.

Active intervention: Although I understand that the HENRY program has been developed and deployed over the past several years in the UK, the reader who is unfamiliar with the current program may benefit from additional information: since the population is mixed. Has this program been tailored to specific racial/ethnic minority groups where dietary practices may be diverse? Since HENRY is a multicomponent program what are the exact elements of this behavioral intervention program? (?motivational interviewing for individual parent/child units, ?supervised group education classes covering a behavioral modification approach toward improving nutrition practices and physical activities’).

Following on from our comment above, the HENRY program can be tailored to specific groups or parents, for example, to align with dietary practices for different ethnicities or religions and this has been clarified within the ‘Active intervention’ section of the paper.

As described in the protocol paper, HENRY uses the Family Partnership model, motivational interviewing and solution focused support. Over 8 sessions, groups of parents practical support in areas such as practical skills in authoritative parenting skills, increasing self-esteem, adopting healthy family lifestyles, goal setting, oral health, active play, portion sizes, and learning about food labels.

Also, a Figure specifying intervention and follow-up timeline would render clarity to how the program will be deployed, and assessments done. For example a parent may register for the program in January 2024, but the actual intervention at that center may not start until another 6 months, if the intervention is over lets say 2-months this child may not have enough time to record an improvement in their BMI compared to another child where the center is well functioning and staff training, intervention occurs promptly and there is enough time to document a treatment effect. A figure with timelines would be helpful. Also, it seems that the intervention is complete after the 8 2-hour sessions, most lifestyle programs have additional booster sessions that improves health education information retention.

Measures are collected 12 months following participant recruitment and recruitment happens within a 6 week window prior to the start of each programme. Thus, there will be minimal differences in the timeline between participants regarding recruitment, intervention delivery and the collection of primary endpoint data.

Data collection: The clinical measures to assess adiposity in children/parents/staff would be weight, height, and waist circumference. Would additional clinical measures to assess general metabolic health such as blood pressure, presence of clinical signs of insulin resistance such as acanthosis in older children be collected? Are there any plans for DEXA scans in a subset of participants?

It seems that no additional biochemical tests are planned in this study. But clinical history such as history of gestational diabetes in the mother, birth weight, breast feeding and others that are established predictors for excess adiposity especially in early life may be additional important data points for this study.

Given the pragmatic approach to this work, this level of additional data collection would not be feasible both in terms of collecting data (participant and researcher burden) and costs of testing. There are therefore no plans to collect additional measures such as blood pressure / DEXA scans. The clinical measures we do collect will provide accurate adiposity data that are invaluable to support individuals. BMI is a proxy measure that is suitable for population comparisons.

Patient recruitment and retention. From reviewing the feasibility study (Ref 1) it is apparent that the study faced barriers with participant recruitment strategies and did not hit their target. The current protocol does not elaborate on recruitment and more importantly retention strategies in this underserved cohort who suffer from many barriers to accessing healthcare.

Our recruitment target for the feasibility was 4 parents per programme and we fell just short of this at 3.9 parents. We explored barriers in our feasibility study and learnt that, while parents indicated that they were keen to take part (determined via consent to contact forms at screening), the research organisation that were responsible for collecting data and registering participants were not able to reach all of these potential participants. We have since started to work with a different organisation (the Clinical Research Network) to expand research data collection coverage and have updated our centre level training to ensure that screening and recruitment is done in a timely way. Further, rather than asking external teams to consent participants, this is done in the centres at the point of screening (now clarified in Section 'Eligibility criteria').

COST ANALYSIS

I commend the authors for their intention to conduct an economic evaluation of an established publicly funded health intervention program that would inform policy makers if the program is cost effective.

Thank you for your detailed consideration of the economic evaluation component of this project.

The authors should provide more details about the methodology of the proposed cost-effectiveness analysis with respect to how the cost-effectiveness acceptability curves will be developed and at what level of willingness to pay (WTP).

The point about WTP is important and of course arises from the challenge we faced that it is not possible to measure (or predict) the health-related quality of life of young infants in any reliable or meaningful way. This explains why we have chosen to report cost-effectiveness in terms of cost per unit change in weight, for which there is no recognised cost-effectiveness willingness-to-pay threshold. We are sorry that in the short text included in the Protocol paper we did not include our full thinking on this matter. Hence we have provided below and in the paper (within 'Economic evaluation') a little more detail based on what we wrote in our original NIHR funding application.

Essentially, the overarching aim of the economic analysis is to reduce decision uncertainty about whether or not HENRY should be commissioned. As we increase the time horizon, from 12 months to 3 years and longer, we anticipate that it is highly likely that decision uncertainty will be reduced in the sense that it will become much clearer whether or not HENRY should be commissioned. This is because the three year and longer-term analyses will incorporate additional health gains and healthcare cost savings that only materialise later on in childhood or early adulthood. If reductions in BMI are observed at 12 months, then we anticipate it will become increasingly likely as we increase

the time horizon that the benefits of HENRY will outweigh the costs of delivering HENRY (all of which are incurred upfront during the 12 months). For example, we might anticipate that the longer term analyses would shift the cost-effectiveness point-estimate from the NE to the SE quadrant of the cost-effectiveness plane. Similarly, we may find that although not cost-saving, the cost per unit change in weight at least falls. On the other hand, if changes in BMI are very small at 12 months, it may become increasingly clear as the time horizon increases that there is little prospect of the intervention being cost-saving. In each of these examples, this would provide stronger evidence to inform commissioning decisions, even if the WTP threshold is unobserved.

In order to develop the CEAC, we would calculate the probability of being cost-effective using a variety of different cost per unit change in weight thresholds, whilst acknowledging that the true WTP threshold is unobserved. The choice of WTP thresholds would be based on existing literature on the cost-effectiveness of comparable interventions and expert opinion from our Project Advisory Group and other networks (e.g. AM is a board member at COBWEB <https://obesitycobweb.wordpress.com/>). We would also present a version of the CEAC which shows the probability of the intervention being cost-saving (regardless of the WTP threshold). In all cases, we will show how the CEAC changes for a given WTP threshold as the analysis time horizon increases. We believe that this is the most meaningful way of presenting our conclusions in order to inform commissioning decisions, without resorting to assumptions about the relationship between BMI and HRQOL at such a young age.

What formal methods of uncertainty analysis using sensitivity/threshold analyses will be done? Will Monte Carlo analysis be done using different types of distribution?

In addition to presenting CEACs for different WTP thresholds described above, uncertainty in our cost-effectiveness estimates will be characterised by presenting bootstrapped estimates of pairs of values for costs and effects on the usual cost-effectiveness plane. One way sensitivity analyses will also be conducted and include assessing the impact of varying the intervention cost and using alternative methods and approaches to matching our trial data with participants in the Millennium Cohort Study (MCS) dataset. Since our analyses all involve regression modelling and trial or secondary data, rather than a microsimulation or Markov-type health economic model where many assumptions would be required about the distributions of key variables and transition probabilities, we opted not to conduct Monte Carlo simulation. The underlying reason for not developing a health economic model of this type is that we are unaware of reliable evidence on the relationship between weight and HRQOL during early childhood. Furthermore, the MCS secondary dataset is a unique and high-quality resource that is well suited to our setting because it enables us to project long-term BMI trajectories and healthcare resource use into early adulthood at the individual-level.

Also, the type of software that will be used should be mentioned.

Stata and/or R will be used for the analysis, depending in part on the preferences of the analyst. We have added version numbers to the manuscript; however, these might change (be updated) by the time of the analysis.

Ref 1. Bryant, M., Collinson, M., Burton, W. et al. Cluster randomised controlled feasibility study of HENRY: a community-based intervention aimed at reducing obesity rates in preschool children. Pilot Feasibility Stud 7, 59 (2021). <https://doi.org/10.1186/s40814-021-00798-z>

Ref 2. Davison KK, Birch LL. Childhood overweight: a contextual model and recommendations for future research. Obes Rev 2001;2: 159–171.

Reviewer: 1

Competing interests of Reviewer: None

Reviewer: 2

Competing interests of Reviewer: n/a

Reviewer: 3

Competing interests of Reviewer: None

VERSION 2 – REVIEW

REVIEWER	Sinha, Madhumita NIH Phoenix
REVIEW RETURNED	01-Mar-2024

GENERAL COMMENTS	Concerns addressed appropriately
----------------------------------

VERSION 2 – AUTHOR RESPONSE